# Robotic Manipulation Learning with Equivariant Descriptor Fields: Generative Modeling, Bi-equivariance, Steerability, and Locality

Jiwoo Kim[*†], Hyunwoo Ryu[*‡], Jongeun Choi[§‡¶] Joohwan Seo[¶], Nikhil Prakash[¶], Ruolin Li[¶], R. Horowitz[¶]
[†]School of Electrical and Electronic Engineering, [‡]Department of Artificial Intelligence,
[§]School of Mechanical Engineering, Yonsei University, Seoul, Republic of Korea
Emails: {nfsshift9801, tomato1mule, jongeunchoi}@yonsei.ac.kr
[¶]Department of Mechanical Engineering, University of California, Berkeley, CA, USA
Emails: {joohwan_seo, nikhilps, ruolin_li, horowitz}@berkeley.edu
[*]Equal Contribution

*Abstract*—Conventional end-to-end visual robotic manipulation learning methods often face challenges related to data inefficiency and limited generalizability. To mitigate these challenges, recent works have proposed incorporating equivariance into their designs. This paper presents a fresh perspective on the design principles of $SE(3)$-equivariant methods for end-to-end visual robotic manipulation learning. Specifically, we examine the recently introduced concept of *Equivariant Descriptor Fields* (EDFs), focusing on four key underlying principles: generative modeling, bi-equivariance, steerable representation, and locality. These principles enable EDFs in achieving impressive data efficiency and out-of-distribution generalizability, even in the absence of prior knowledge. By comparing EDFs with other contemporary equivariant methods based on the four criteria, this paper underscores the importance of these design principles and aims to establish a guiding framework for future research on $SE(3)$-equivariant robotic manipulation.

## I. INTRODUCTION

Recently, equivariant methods have gained notable attention due to their data efficiency, robustness and generalizability. Incorporating equivariance has shown promising results in various fields, including protein [15, 11], molecule [12, 4], 3D object segmentation [17, 6], shape reconstruction [1, 2], and reinforcement learning [27, 18, 31].

For learning manipulation tasks, the prerequisite for numerous demonstrations and rollouts [8, 14, 7, 39, 16] is a critical weakness. Recent works reveal that incorporating equivariance can improve data efficiency and generalizability. The $SE(2)$-*equivariance* (planar roto-translation equivariance) has been used to improve the efficiency of behavior cloning [40, 13, 21] and reinforcement learning methods [30, 28, 29, 41] for planar tasks. For highly spatial tasks, the $SE(3)$-equivariance (spatial roto-translation equivariance) is required. Neural Descriptor Fields (NDFs) [23] and their variants [24, 3] leverage this property to achieve remarkable data efficiency and generalizability. However, they cannot be end-to-end trained; instead, they require pre-training and object segmentations.

To overcome this challenge, *Equivariant Descriptor Fields* (EDFs) [20] has been proposed. EDFs are end-to-end trainable models for $SE(3)$-equivariant visual manipulation learning. Different from previous $SE(3)$-equivariant methods, EDFs are capable of learning manipulation tasks from only a few demonstrations without requiring any prior knowledge, such as pre-training and object segmentation.

In this paper, we examine the four key design principles of EDFs and compare them with other recent works. By doing so, we seek to offer a novel perspective that can pave the way for subsequent studies on equivariant methods for robotic manipulation learning.

## II. PRELIMINARIES: REPRESENTATION THEORY

A representation $D$ is a map from a group $\mathcal{G}$ to an invertible matrix $GL(N) \in \mathbb{R}^{N \times N}$ that satisfies $D(g)D(h) = D(gh)$ for every $g, h \in \mathcal{G}$. In particular, any representation of $SO(3)$ can be expressed as a block-diagonal matrix composed of *real Wigner D-matrices* by a change of basis. A real Wigner D-matrix $D_l(R) \in \mathbb{R}^{(2l+1) \times (2l+1)}$ of degree $l \in \{0, 1, 2, ...\}$ are orthogonal matrices that are *irreducible*, meaning that they cannot be block-diagonalized anymore. Therefore, Wigner D-matrices constitute the building blocks of any representations of $SO(3)$. A *type-l vector* is a $(2l + 1)$-dimensional vector that is transformed by $D_l(R)$ under rotation $R \in SO(3)$. Type-0 vectors are invariant to rotations (i.e. scalars) such that $D_0(R) = I$. On the other hand, type-1 vectors are rotated according to the 3D rotation matrices, that is, $D_1(R) = R$.

Let $\mathcal{O}$ be the set of all possible colored point clouds. A point cloud is given by $O = \{(x_i, c_i) : i \in \mathcal{I}\}$, where $x_i \in \mathbb{R}^3$ and $c_i \in \mathbb{R}^3$ are point $i$'s position and color. A type-$l$ vector field $f : \mathbb{R}^3 \times \mathcal{O} \to \mathbb{R}^{2l+1}$ generated by $O \in \mathcal{O}$ is $SE(3)$-equivariant if $D_l(R)f(x|O) = f(gx|g \cdot O)$, $\forall g = (p, R) \in SE(3)$, $x, p \in \mathbb{R}^3$, $O \in \mathcal{O}$ and $g \cdot O = \{(gx_i, c_i) : i \in \mathcal{I}\}$.

## III. EQUIVARIANT DESCRIPTOR FIELDS: THE FOUR KEY MODEL PROPERTIES

In what follows, we will delve into EDFs and compare them with other equivariant models, focusing on the four key principles, viz., *generative modeling*, *bi-equivariance*, *steerable representation* and *locality* (see Table I).

TABLE I: Comparison of recently proposed equivariant methods for robotic manipulation learning.

| Method | Bi-Equivariance | | Locality | Steerable Representations | Generative Modeling | End-to-end Training |
|---|---|---|---|---|---|---|
| | Left Equiv. | Right Equiv. | | | | |
| Transporter Networks [40] | $SE(2)$ | Translation | ◯ | Invariant | ✕ | ◯ |
| Equivariant Transporter Networks [13] | $SE(2)$ | $SE(2)$ | ◯ | **Equivariant** | ✕ | ◯ |
| Equivariant RL (SAC/DQN) [28, 29, 30] | $SE(2)$ | $\mathbb{Z}_2$ | ◯ | **Equivariant** | ✕ | ◯ |
| NDFs [23] | $\boldsymbol{SE(3)}$ | ✕ | ✕ | Invariant | ✕ | ✕ |
| L-NDFs [3] | $\boldsymbol{SE(3)}$ | ✕ | ◯ | Invariant | ✕ | ✕ |
| R-NDFs [24] | $\boldsymbol{SE(3)}$ | $\boldsymbol{SE(3)}$ | ✕ | Invariant | ✕ | ✕ |
| **EDFs** [20] | $\boldsymbol{SE(3)}$ | $\boldsymbol{SE(3)}$ | ◯ | **Equivariant** | ◯ | ◯ |

## A. Generative Modeling

In practice, expert demonstration policies for robotic manipulation tasks are rarely unimodal. To illustrate this, consider a mug-picking task. The human expert may occasionally choose to grasp the mug by the rim and at other times by the handle. To properly learn such multimodalities, generative modeling is required for the policy distributions [19] (see Fig. 1). As shown in Fig. 1, naively regressing or discretizing the policy results in suboptimal policy distributions. On the other hand, generative models such as energy-based models (EBMs) and diffusion models capture the behavior more accurately. EDFs utilize EBMs' approach to model the policy distribution, enabling both end-to-end training and sampling. This is in contrast to the energy minimization method used by NDFs variants [23, 24, 3], which requires frozen pre-trained networks.

The EDFs' energy-based policy conditioned by the point cloud observations of the scene $O^{scene}$ and the grasped object $O^{grasp}$ is defined on the $SE(3)$ manifold as

$$P(g|O^{scene}, O^{grasp}) = \frac{\exp[-E(g|O^{scene}, O^{grasp})]}{Z}$$
$$\text{where } Z = \int_{SE(3)} dg \exp[-E(g|O^{scene}, O^{grasp})], \quad (1)$$

where $E$ is an energy function which will be defined later.

## B. Bi-equivariance

To successfully perform object picking tasks, it is crucial for the end-effector pose to be equivariant to changes in the initial pose of the target object within the *scene*. To illustrate this *scene equivariance*, consider a task in which the end-effector pose $g_{WE} \in SE(3)$ in the world frame $W$ should be inferred from the observation of the scene $O^{scene}$. Here, $g_{WE} := (p_{WE}, R_{WE}) \in SE(3)$ denotes the specification of the configuration of the end-effector frame $E$ relative to $W$. Now, consider a new world frame $W'$. The reference frame change $\Delta g_W = g_{W'W} \in SE(3)$ induces the following transformations in the scene observation and end-effector pose.

$$O_{W'}^{scene} = \Delta g_W \cdot O_W^{scene} = g_{W'W} \cdot O_W^{scene}$$
$$g_{W'E} = \Delta g_W \, g_{WE} = g_{W'W} g_{WE}$$

The corresponding equivariant probabilistic policy[1] $P$ against $\Delta g$ then must satisfy

$$P(\Delta g_W \, g_{WE} | \Delta g_W \cdot O_W^{scene}) = P(g_{W'E} | O_{W'}^{scene})$$

[1] The equivariant probabilistic policy implies *invariance* of the conditional probabilities when the state and action are *equivariantly* transformed

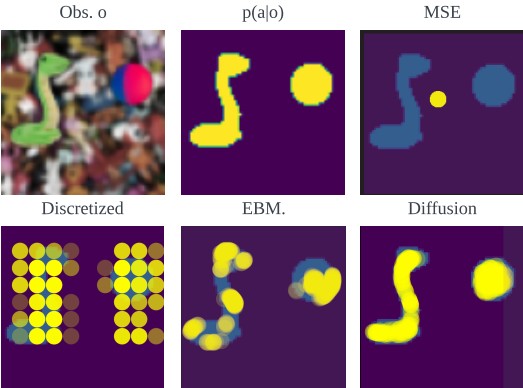

Fig. 1: Comparison of behavior cloning methods: Generative models (EBM and Diffusion) accurately capture multimodal behaviors of the oracle policy $p(a|o)$ compared to regression (MSE) or discretized methods. Reproduced with authors' permission [19].

Since the perturbation $\Delta g_W$ appears on the left side of $g$, we refer to this scene equivariance as *left equivariance*. We illustrate left equivariance in Fig. 2.

However, as it turns out, left equivariance alone is insufficient to successfully perform object placing tasks. Unlike picking tasks, which only require observing the scene, placing tasks also requires the observation of the grasp, which adds another layer of complexity to the problem. Furthermore, the grasp pose inferred by a pick policy learned from a few expert demonstrations may not be optimal. As a result, the grasped object may be in a pose that has never been shown by the expert demonstrations. Hence, object placing tasks require another type of equivariance, namely the *grasp equivariance*. Consider the same object pose $B$ being grasped in two different manners, respectively $E$ and $E'$. Let $O_E^{grasp}$ be the observation of the object grasped by an end-effector with frame $E$. We assume that frame $B$ is attached to the grasped object such that $g_{EB}$ is the pose of $B$ relative to frame $E$. A transformation of the grasped object pose due to a change $\Delta g$ between end-effector frames $E$ and $E'$, as shown in Fig. 3, induces the transformed observation relative to frame $E'$:

$$O_{E'}^{grasp} = \Delta g_E \cdot O_E^{grasp} = g_{E'E} \cdot O_E^{grasp}.$$

To keep the relative pose between the scene and the grasped object invariant for equivariance of the probabilistic policy, the end-effector pose must be transformed by $\Delta g_E$ such that

$$g_{WB} = g_{WE}g_{EB} = g_{WE'}g_{E'B} = g_{WE'}g_{E'E}\,g_{EB} = g_{WE'}\Delta g_E\, g_{EB}$$
$$\Rightarrow g_{WE'} = g_{WE}\Delta g_E^{-1}$$

A probabilistic policy $P$ under such an equivariance requires

$$P(g_{WE}\Delta g_E^{-1}|O_E^{scene}, \Delta g_E \cdot O_E^{grasp}) = P(g_{WE'}|O_{E'}^{scene}, O_{E'}^{grasp})$$

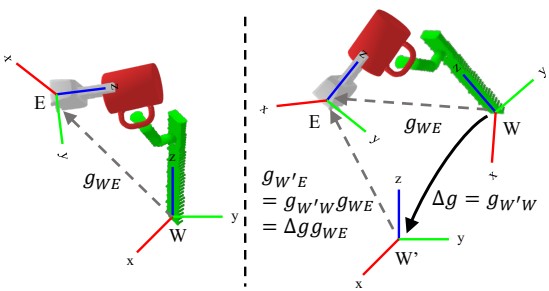

Fig. 2: The left equivariance illustrates that the target pose is equivariant to the transformation of the *scene*, as such the perturbation $\Delta g$ is on the left of $g$.

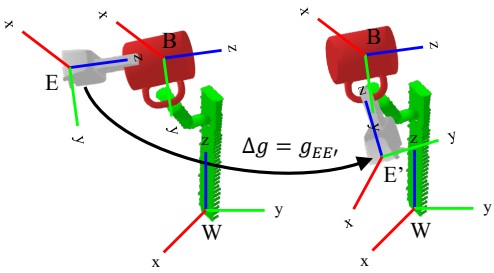

Fig. 3: The right equivariance implies that the target pose is equivariant to the *grasp* state, in which the perturbation $\Delta g$ is located on the right of the $g$.

Notice that such a grasp equivariance is a *right equivariance* since the inverse of the perturbation $\Delta g_E^{-1}$ appears on the right side of $g$. We illustrate the right equivariance in Fig. 3. Combining both the left and right equivariances, we finally define *bi-equivariance* [20] as follows.

$$
\begin{aligned}
P(g|O^{scene}, O^{grasp}) &= P(\Delta g_W g | \Delta g_W \cdot O^{scene}, O^{grasp}) \\
&= P(g\Delta g_E^{-1}|O^{scene}, \Delta g_E \cdot O^{grasp})
\end{aligned}
\quad (2)
$$

Among $SE(2)$-equivariant methods, Transporter Networks [40] and recently proposed equivariant reinforcement learning methods [28, 29, 30] are left equivariant, but not fully right equivariant (only translation equivariant). On the other hand, Equivariant Transporter Networks [13] incorporate full $SE(2)$ bi-equivariance, thereby achieving significant increase in data efficiency over Transporter Networks. Among $SE(3)$-equivariant methods, Neural Descriptor Fields (NDFs) [23] and Local Neural Descriptor Fields (L-NDFs) [3] are uni-equivariant methods. Since NDFs and L-NDFs assume a fixed placement target pose, bi-equivariance is not required. However, to solve more general tasks such as object rearrangement tasks, bi-equivariance becomes essential. Relational Neural Descriptor Fields (R-NDFs) [24] are a bi-equivariant method for object rearrangement tasks. However, pre-trained NDFs and a human annotated object keypoint are required to equivariantly align query points for the training.

On the other hand, EDFs [20] directly infer query points using an $SE(3)$-equivariant query density model that can be end-to-end trained. EDFs achieve bi-equivariance for the policy in (1) with a bi-equivariant energy function $E(g|O^{scene}, O^{grasp})$. The specific design of this energy function will be introduced subsequently.

### C. Steerable Representation

To achieve robust equivariant manipulation, a model must utilize symmetric feature representations from the observations. Steerable representations are proficient in representing these features due to their orientation sensitivity [33] (see Fig. 4). Moreover, due to continuous expressions, steerable representations acquire rigorous information compared to the discretization methods and demonstrate better precision as evidenced by [1].

Importantly, compared to rotation invariant features, steerable features are superior in encoding the orientations of local geometries. To encode orientation information using rotation invariant features, they must be spatially distributed, breaking

locality. For example, the color vector (red, green, blue) is such a rotation invariant feature. To determine the rigid-body orientation, at least three non-collinear points of different colors are required. Conversely, one can represent orientation with only a single point, using rotation equivariant, or steerable features. Thus, orientation information can be localized into a single point, better capturing the local geometry. This makes the learned features more generalizable and less sensitive to disturbances.

Transporter Networks [40] and Neural Descriptor Fields variants [23, 24, 3] utilize rotation invariant feature fields to obtain equivariance (e.g., Feature map of CNNs can be thought of as 2-dimensional feature fields). Alternatively, Huang et al. [13], Wang et al. [30, 28, 29] utilize the steerable features of the $C_n$ group (discretized $SO(2)$ group), thereby significantly improving data efficiency.

An EDF $\varphi(x|O)$ is defined as the concatenation of $N$ $SO(3)$-steerable vector fields that are $SE(3)$-equivariant

$$
\varphi(x|O) = \bigoplus_{n=1}^{N} \varphi^{(n)}(x|O)
$$

where $\varphi^{(n)}(x|O) : \mathbb{R}^3 \times \mathcal{O} \to \mathbb{R}^{2l_n+1}$ is an $SE(3)$-equivariant *type-$l_n$* vector field generated by $O$. Therefore, $\varphi(x|O)$ is transformed according to $g = (p, R) \in SE(3)$ as

$$
\begin{aligned}
\varphi(gx|g \cdot O) &= D(R)\varphi(x|O) \\
&= \begin{pmatrix} D_{l_1}(R) & \cdots & \emptyset \\ \vdots & \ddots & \vdots \\ \emptyset & \cdots & D_{l_n}(R) \end{pmatrix} \varphi(x|O)
\end{aligned}
$$

where $D(R)$ is a block diagonal of Wigner D-matrices.

The $SE(3)$ bi-equivariant energy function for the EBM in Eq. (1) can be constructed with EDFs as

$$
E(g|O^{scene}, O^{grasp}) =
$$
$$
\int_{\mathbb{R}^3} d^3x\, \rho(x|O^{grasp})\|\varphi(gx|O^{scene}) - D(R)\psi(x|O^{grasp})\|^2
\quad (3)
$$

where $\varphi(x|O^{scene})$ is the *key EDF*, $\psi(x|O^{grasp})$ is the *query EDF*, and the $\rho(x|O^{grasp})$ is the *query density*, which are all $SE(3)$-equivariant and learnable neural fields.

### D. Locality

For a robotic manipulation model to be robust, it must be able to pick and place objects in previously unseen poses.

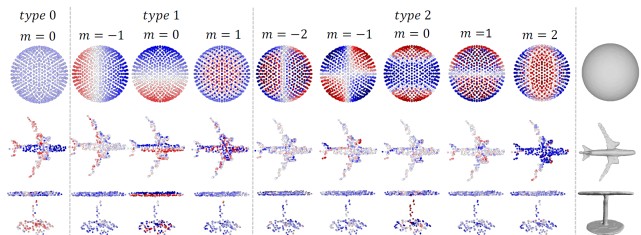

| *type 0* | *type 1* | | | *type 2* | | | |
|---|---|---|---|---|---|---|---|
| $m = 0$ | $m = -1$ | $m = 0$ | $m = 1$ | $m = -2$ | $m = -1$ | $m = 0$ | $m = 1$ | $m = 2$ |

Fig. 4: Visualization of *type-l features* ($l = 0, 1, 2, …$) for a sphere (top), airplane (middle), and table (bottom). Higher-type features are sensitive to the orientations of local geometries such as planes and corners. Reproduced with the authors' permission [1].

If the model can learn local geometric structures that are shared across different objects, it would greatly increase its generalizability. For example, if a model was trained to pick a mug by holding the rim, the similarities in the local geometric features can be utilized to grasp other objects by the rim. Consequently, locality is critical for generalizability and data efficiency. Recent studies in various fields such as robotics [3], point cloud segmentation [6], and shape reconstruction [1] highlight the importance of incorporating locality in equivariant methods.

Another benefit of imposing locality to equivariant methods is that the target object does not require to be segmented from the backgrounds. For unsegmented observations, only equivariance to the target object is desired, and the equivariance to backgrounds must be suppressed. We name this property as *local equivariance*, in contrast to *global equivariance* (see Fig. 5). However, naively applying Eq. (2) can only guarantee global equivariance. Therefore, special care must be taken in designing methods to respect the locality of the tasks so as to obtain local equivariance.

For example, Transporter networks and their variants [40, 21, 13] naturally exploit the locality of convolutional neural networks. Therefore, Transporter Networks and their variants can be used without object segmentation pipelines or any other object centric assumptions. On the other hand, NDFs [23] and R-NDFs [24] rely on centroid subtraction methods to achieve translational equivariance. Due to the highly non-local nature of centroid subtraction, these methods require the target object to be segmented from the background.

EDFs utilize a *Tensor Field Network (TFN)* [25] model for the final layer and $SE(3)$-*transformers* [10] in other layers. These methods rely on spatial convolutions, enabling the easy acquisition of locality by using convolution kernels with finite support. This is in contrast to the Vector Neurons [5] method that were used for NDFs and R-NDFs.

We provide more details on the training, sampling, and the implementation details in Appendix A. Mathematical proofs can be found in the original paper of EDFs [20].

## IV. EXPERIMENTAL RESULTS

To evaluate the EDFs' generalization performance with other methods, Ryu et al. [20] conducted experiments with a mug-hanging task and a bowl/bottle pick-and-place task. The objective is to pick a mug or bowl/bottle and place it on a randomly posed hanger or plate. For the evaluation, multiple

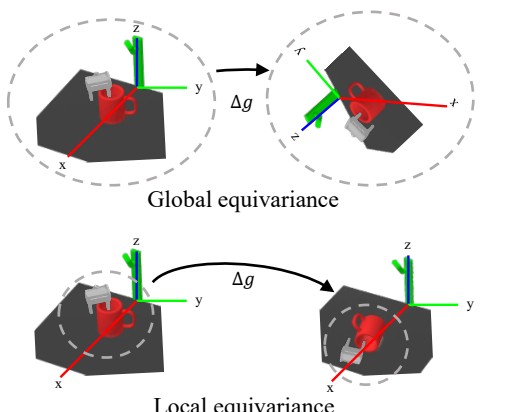

Global equivariance

Local equivariance

Fig. 5: The difference of global equivariance and local equivariance. The global equivariance represents the translation of the whole scene, while the local equivariance denotes the translation of the target object.

scenarios including unseen poses, unseen distracting objects, and unseen instances in randomized poses were used.

First, EDFs were compared with $SE(3)$ Transporter Networks [40], which are the extensions to the original Transporter Networks that regress additional degrees of freedom (height, roll, pitch). Table II of Appendix B shows that EDFs out-preform Transporter Networks in all three tasks. By comparing the results, EDFs turn out to be more robust than Transformer Networks, illustrating the significance of the $SE(3)$-equivariance when it comes to highly spatial tasks.

In comparing EDFs to NDFs [23], it was necessary to account for some of NDFs' limitations such as the fact that NDFs require segmentations and a fixed pose of the placement target. Thus, EDFs were compared against an NDF-like constructed baseline model, which uses only the type-0 descriptor features. From Table III of Appendix B, EDFs, which use higher type descriptors, surpass the performance of the NDF-like model. Additional experimental descriptions and results can be found in Appendix B and the original paper [20].

## V. CONCLUSION

We introduce EDFs and emphasize the importance of the following four properties: 1) generative modeling, 2) bi-equivariance, 3) steerable representations, and 4) locality; in order to synthesize noteworthy equivariant robotic manipulation learning models. We demonstrate the effectiveness and the generalization of EDFs in inferring the target pose in spite of previously unseen instances, unseen poses, and distracting objects using only a few demonstrations.

For future research, it could be beneficial to integrate $SE(3)$-equivariant shape reconstruction and SLAM methods [38, 1, 2, 9] with EDFs to overcome incomplete and noisy point cloud observations. Expanding EDFs to trajectory-level problem is also an important issue. For kinematic and dynamic trajectory planning, one might consider incorporating guided diffusion methods [26] and geometric impedance control framework [22] respectively. Lastly, to improve the speed of the MCMC sampling required for EDFs, techniques such as amortized sampling [36, 32] and cooperative learning [34, 35, 37] could be explored.

ACKNOWLEDGMENTS

This work was supported by the National Research Foundation of Korea (NRF) grants funded by the Korea government (MSIT) (No.RS-2023-00221762 and No. 2021R1A2B5B01002620). This work was also partially supported by the Korea Institute of Science and Technology (KIST) intramural grants (2E31570).

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

## Appendix

### A. Equivariant Descriptor Fields

For a thorough understanding of the EDFs [20], we reproduce the training, sampling, and implementation details in this section. We denote the learnable parameters as $\theta$. Further details and proofs can be found in the original paper [20]. The overview of the methodology is illustrated in Fig. 6.

*a) Training:* For the training of the energy-based model Eq.1, the gradient of the log-likelihood at the demonstrated target end-effector pose $g_{target}$ is be approximated as

$$\nabla_\theta \log P_\theta(g_{target}|O^{scene}, O^{grasp}) \approx \\ - \nabla_\theta E_\theta(g_{target}|O^{scene}, O^{grasp}) \\ + \frac{1}{N}\sum_{n=1}^{N}[\nabla_\theta E_\theta(g_n|O^{scene}, O^{grasp})]$$

where $g_n \sim P_\theta(g_n|O^{scene}, O^{grasp})$ is the *n*-th negative sample, which is sampled from the model.

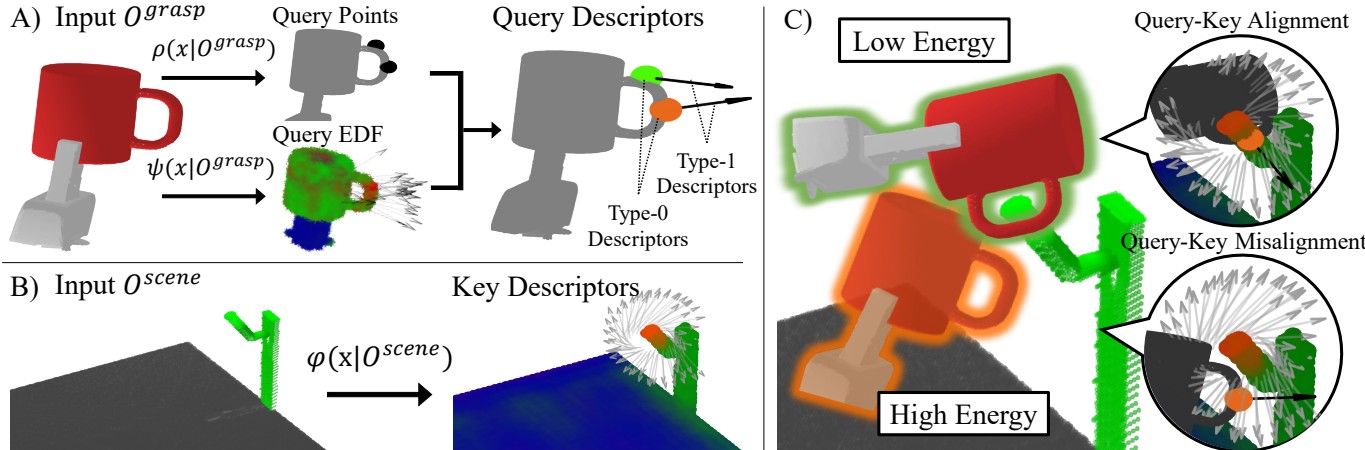

Fig. 6: A) The query points and the query EDF are generated from the grasp point cloud $O^{grasp}$. Each query point is assigned with the corresponding query descriptor, which is the field values of the query EDF at the query points. The type-0 descriptors are visualized as colors and type-1 descriptors as arrows. The higher descriptors are not visualized. B) Similarly, the key EDFs are generated from $O^{scene}$. C) The query descriptors are transformed and matched to the key descriptors to produce the energy value. As shown in the visualization, the lower energy case has a better alignment of the query and the key descriptors, while the high energy case fails to do so. MCMC methods are used to sample end-effector configurations according to their energy (lower energy means exponentially higher probability). Reproduced and modified with the authors' permission [20].

*b) Sampling:* Energy-based models typically do not allow direct sampling. Therefore, Ryu et al. [20] utilize Monte-Carlo Markov Chain (MCMC) methods to sample end-effector poses from Eq.1. In particular, two-stage sampling strategy is used. First, the Metropolis-Hastings algorithm(MH) is used to rapid explore the workspace. Next, the Langevin dynamics on the $SE(3)$ manifold is employed. The samples gained from MH are used as the initial seeds for the Langevin dynamics. In the quaternion-translation parametrization, the differential equation for the langevin dynamics on the $SE(3)$ manifold is

$$dz = \begin{pmatrix} d\hat{h} \\ dv \end{pmatrix} = -LL^T \nabla_z E(z)\, dt + \sqrt{2}L\, dw$$

$$L = \begin{bmatrix} L_{SO(3)} & 0_{4\times 3} \\ 0_{3\times 3} & I_{3\times 3} \end{bmatrix}$$

$$L_{SO(3)} = \frac{1}{2} \begin{bmatrix} -h^2 & -h^3 & -h^4 \\ h^1 & -h^4 & h^3 \\ h^4 & h^1 & -h^2 \\ -h^3 & h^2 & h^1 \end{bmatrix}$$

where $z = (\hat{h}, v) \cong S^3 \times \mathbb{R}^3 \subset \mathbb{R}^7$ is the quaternion-translation parameterization of $SE(3)$ with $\hat{h} = h^1 + h^2\,\hat{i} + h^3\,\hat{j} + h^4\,\hat{k}$.

*c) Implementation:* As mentioned in Section III-D, Ryu et al. [20] employed the $SE(3)$-Transformers [10] and Tensor Field Networks (TFNs) [25] for the implementation of the two EDFs $\varphi(x|O^{scene})$ and $\psi(x|O^{grasp})$ in Eq. 3. For the tractability of the integral in Eq. 3, Ryu et al. [20] modeled the equivariant query density field $\rho(x|O^{grasp})$ as weighted sum of query points $Q \in \mathbb{R}^{N_Q \times 3}$ such that

$$\rho_\theta(x|O^{grasp}) = \sum_{i=1}^{N_Q} \left[ w_\theta\left(x|O^{grasp}\right) \delta^{(3)}\left(x - Q_{i;\theta}\left(O^{grasp}\right)\right) \right] \tag{4}$$

where $Q_{i;\theta}\left(O^{grasp}\right)$ is the *query point model* that infers the position of the $i$-th query point from $O^{grasp}$, and $w_\theta\left(x|O^{grasp}\right)$ is an equivariant scalar field that bestows the

weight to each query points. Here, $\delta^{(3)} = \prod_{i=1}^{3} \delta(x_i)$ denotes the Dirac-delta function on $\mathbb{R}^3$. Instead of using separate model for $Q_{i;\theta}(O^{grasp})$, Ryu et al. [20] used Stein Variational Gradient Descent (SVGD) method to equivariantly draw query points from $w_\theta(\cdot|O)$. Note that $w_\theta(\cdot|O)$ can be considered as a special case of EDFs with only a single type-0 descriptor. Therefore, $SE(3)$-Transformers and TFNs can be utilized for the implementation.

With Eq. 4, the integral in the energy function Eq. 3 can be written as a tractable summation form as follows

$$E_\theta(g|O^{scene}, O^{grasp})$$
$$= \sum_{i=1}^{N_Q} w_{i;\theta} \left\| \varphi_\theta(g\, Q_{i;\theta}|O^{scene}) - D(R)\psi_\theta(Q_{i;\theta}|O^{grasp}) \right\|^2$$

where $Q_{i;\theta} = Q_{i;\theta}\left(O^{grasp}\right)$ and $w_{i;\theta} = w_{i;\theta}\left(Q_{i;\theta}|O^{grasp}\right)$.

### B. Experimental Results

This section reproduces the details on the experiments from Ryu et al. [20] that were conducted to compare EDFs with prior methods. The mug-hanging, and bowl/bottle pick and place tasks were employed for comparison. The models were trained with ten demonstrations for each task where the cup, bowl, and bottle were positioned upright as shown in Fig. 7. For evaluation, the models were given various scenes with an unseen instance, in a random posture, with various distracting objects nearby, as shown in Fig. 8.

First, Table II compares EDFs with the state-of-the-art end-to-end visual manipulation method, Transporter Networks [40]. Specifically, the $SE(3)$-extended version of the original Transporter Networks ($SE(3)$-TNs) proposed in [40] is used. $SE(3)$-TNs directly regress the additional three degrees of freedom (height, roll, pitch) of the planar Transporter Networks. Therefore, despite its name, $SE(3)$-TNs are $SE(2)$-equivariant methods. For each of the three tasks, four different scenarios were tested: 1) the target object is an unseen target instance, 2) the target instance is positioned in a random

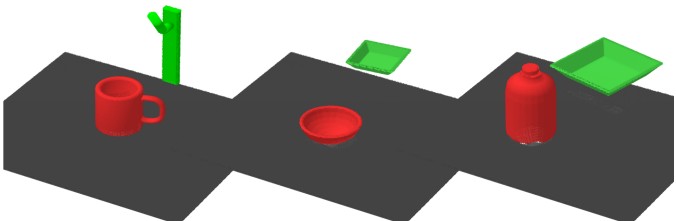

Fig. 7: The scenes that are used to train the methods. For each demonstration, there are either a cup, bowl, or bottle pose only upright in random locations. Reproduced with the authors' permission [20].

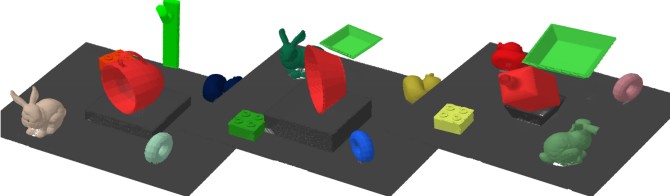

Fig. 8: The scenarios that are given to evaluate the models. New instances are given that were not seen during training, and they are positioned in random postures. In addition, there are several distracting objects around the target instance. Reproduced with the authors' permission [20].

orientation, 3) the target instance is surrounded by various unseen distracting objects, and finally 4) all of the three unseen conditions are combined.

As can be seen in Table II, EDFs significantly outperform Transporter Networks in all of the four unseen scenarios. Especially, Transporter Networks completely fail when the target object is provided in previously unseen poses (Scenario 1), due to the lack of the spatial $SE(3)$-equivariance. For example, as shown in Fig. 9-A, Transporter Networks fail to pick the target instance when positioned in an unseen pose and anticipate to grab the instance as if it were positioned upright as it was during training. On the other hand, EDFs successfully infer appropriate end-effector poses in all of the cases, evidencing the importance of the $SE(3)$ bi-equivariant modeling.

Next, Ryu et al. [20] conducted another experiment to validate the importance of steerable representations. For the comparison, an ablated model without steerable representations that is analogous to NDFs variants [23, 24, 3] was used. Notably, unlike these previous works [23, 24, 3], this experiment did not use category-level pre-training, necessitating greater generalization capabilities for the model to successfully pick-and-place unseen object instances. The results are summarized in Table III. The ablated model utilizes only the type-0 descriptors, which are invariant to rotations. Therefore, as illustrated in Fig.9-B, the ablated method struggles to correctly infer the orientations of the target poses for previously unseen instances. In contrast, EDFs utilize higher descriptors, hence are capable of accurately inferring the target poses. The experimental results show that steerable representations are crucial for improving the orientational accuracy and generalizability of inferred pick-and-place poses.

Lastly, Ryu et al. [20] conducted an experiment to assess the robustness of EDFs under significant multimodality in the demonstrations. In this experiment, EDFs were trained with three different demonstration sets for mug-hanging task:

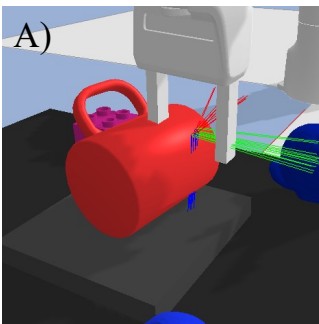
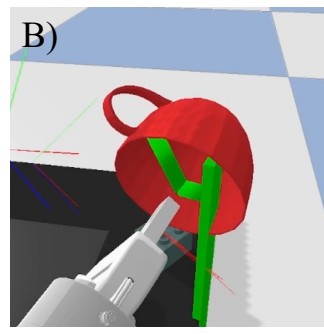

Fig. 9: A) Transformer Networks exhibit the inability to pick the target instance that is posed in an unseen posture due to their lack of $SE(3)$-equivarince. B) NDF-like models, which only use the type-0 descriptors, fail to place the cup on the hanger due to the lack of orientational sensitivity of the target instance. Reproduced with the authors' permission [20].

1) unimodal, low-variance demonstrations (only picking a specific point on the mug), 2) diverse but consistent demonstrations (multimodal, but always picks by the rim of the mug), and 3) diverse and inconsistent demonstrations (multimodal, picking the mug by either the rim or the handle). The results are summarized in Table IV. Comparing the results of training demonstration set 1 (unimodal) and 2 (multimodal and consistent), we observe that EDFs are robust to the multimodality in the demonstrations. Furthermore, the experimental results suggest that EDFs actually benefit from the diversity of multimodal demonstrations. This can be attributed to the nature of generative models, that are flexible enough to leverage diverse pick-and-place strategies. Moreover, this generative nature of EDFs allows them to be tolerable to highly inconsistent demonstrations. As can be seen in the results for demonstration set 3 (multimodal and inconsistent), EDFs are shown to be robust to inconsistency in the demonstrations.

These comprehensive experiments reveal the importance of the four criteria in designing equivariant methods for end-to-end visual robotic manipulation. Further experimental results and explanations can be found in the original paper [20].

TABLE II: Pick-and-place success rates in various out-of-distribution settings. Reproduced with authors' permission [20].

| | Mug | | | Bowl | | | Bottle | | |
|---|---|---|---|---|---|---|---|---|---|
| | Pick | Place | Total | Pick | Place | Total | Pick | Place | Total |
| **Unseen Instances** | | | | | | | | | |
| $SE(3)$-TNs [40] | 1.00 | 0.36 | 0.36 | 0.76 | 1.00 | 0.76 | 0.20 | 1.00 | 0.20 |
| EDFs (Ours) | 1.00 | **0.97** | **0.97** | **0.98** | 1.00 | **0.98** | **1.00** | 1.00 | **1.00** |
| **Unseen Poses** | | | | | | | | | |
| $SE(3)$-TNs [40] | 0.00 | N/A | 0.00 | 0.00 | N/A | 0.00 | 0.00 | N/A | 0.00 |
| EDFs (Ours) | **1.00** | 1.00 | **1.00** | **1.00** | 1.00 | **1.00** | **0.95** | 1.00 | **0.95** |
| **Unseen Distracting Objects** | | | | | | | | | |
| $SE(3)$-TNs [40] | 1.00 | 0.63 | 0.63 | 1.00 | 1.00 | 1.00 | 0.96 | 0.92 | 0.88 |
| EDFs (Ours) | 1.00 | **0.98** | **0.98** | 1.00 | 1.00 | 1.00 | **0.99** | **1.00** | **0.99** |
| **Unseen Instances, Arbitrary Poses & Distracting Objects** | | | | | | | | | |
| $SE(3)$-TNs [40] | 0.25 | 0.04 | 0.01 | 0.09 | 1.00 | 0.09 | 0.26 | 0.88 | 0.23 |
| EDFs (Ours) | **1.00** | **0.95** | **0.95** | **0.95** | 1.00 | **0.95** | **0.95** | **1.00** | **0.95** |

TABLE III: Success rate and inference time of the ablated model and EDFs. All the evaluations are done in the *unseen instances, poses & distracting objects* setting. Reproduced with authors' permission [20].

| | Mug | | | Bowl | | | Bottle | | |
|---|---|---|---|---|---|---|---|---|---|
| Descriptor Type | Pick | Place | Total | Pick | Place | Total | Pick | Place | Total |
| **NDF-like (Type-0 Only)** | | | | | | | | | |
| Inference Time | 5.7s | 8.6s | 14.3s | 6.1s | 9.9s | 16.0s | 5.8s | 17.3s | 23.0s |
| Success Rate | 0.84 | 0.77 | 0.65 | 0.60 | 0.95 | 0.57 | 0.66 | 0.95 | 0.63 |
| **EDFs (Type-0∼3)** | | | | | | | | | |
| Inference Time | 5.1s | 8.3s | 13.4s | 5.2s | 10.4s | 15.6s | 5.2s | 11.5s | 16.7s |
| Success Rate | **1.00** | **0.95** | **0.95** | **0.95** | **1.00** | **0.95** | **0.95** | **1.00** | **0.95** |

TABLE IV: Success rate of EDFs for mug-hanging task with different demonstrations. Reproduced with authors' permission [20].

| | Low Var. & Unimodal Grasps | | | Diverse and Consistent Grasps (Rim Only) | | | Diverse and Inconsistent Grasps (Handle & Rim) | | |
|---|---|---|---|---|---|---|---|---|---|
| Setup | Pick | Place | Total | Pick | Place | Total | Pick | Place | Total |
| Unseen Poses (P) | 1.00 | 0.96 | 0.96 | 1.00 | 1.00 | 1.00 | 1.00 | 0.99 | 0.99 |
| Unseen Instances (I) | 0.99 | 0.90 | 0.89 | 1.00 | 0.97 | 0.97 | 1.00 | 0.92 | 0.92 |
| Unseen Distractors (D) | 1.00 | 1.00 | 1.00 | 1.00 | 0.98 | 0.98 | 0.96 | 0.99 | 0.95 |
| Unseen P+I+D | 0.99 | 0.83 | 0.82 | 1.00 | 0.95 | 0.95 | 0.90 | 0.89 | 0.80 |