# OpenReview forum: "Robotic Manipulation Learning with Equivariant Descriptor Fields: Generative Modeling, Bi-equivariance, Steerability, and Locality"
_roboticsfoundation.org/RSS/2023/Workshop/Symmetry — RSS 2023 Workshop Symmetry Oral_

### Official Review · Reviewer_pcN6 · 2023-06-11
**Review by reviewer pcN6**

**Rating:** 9
**Confidence:** 5

**Review:**

The paper presents Equivariant Descriptor Fields, an SE(3)-equivariant learning method for robotic manipulation. This method stands out from previous works in equivariant manipulation learning as it achieves all of bi-equivariance, locality, equivariant representations, generative modeling, and end-to-end training. The paper not only demonstrates a novel method with impressive experimental results, but also provides a comprehensive overview of generative modeling, bi-equivariance, steerable representation, and locality, which could be greatly beneficial to the community. However, the extensive analysis also constitutes the shortcoming of the paper — the insufficient explanation of the EDF method. I acknowledge that this is partially because of the 4-page limit of the workshop, but I believe that a more detailed method description could strengthen the paper. I suggest that the author relocate a portion of the analysis of Generative Modeling, Steerable Representation, and Locality to the appendix, and include a clearer description of the EDF algorithm (especially a high-level step-by-step explanation).

---

### Official Review · Reviewer_i7yZ · 2023-06-19

**Rating:** 7
**Confidence:** 4

**Review:**

**Summary**

This work proposed an extended version of Equiavarinat Descriptor Fields (EDF) [20], a novel approach enabling end-to-end visual robotic manipulation learning with SE(3) equivariance, that’s coupled with four key desirable properties: generative modeling, bi-equivalence, steerable representation, and locality. This work adopts, improves, and connects many existing techniques from previous works to enable a system with the above-mentioned desirable properties. However, the writing is not self-contained as a lot of the details are to be found in a published earlier work, [20]. The difference between this work and [20] could be made more explicit in writing. In addition, the true extent of the claimed advantages brought by the proposed system could be further supported with real-world robot experiments.

**Strengths**
1. It tried to tackle an important and well-motivated problem of enabling end-to-end visual robot manipulation learning with SE(3)-equivariance.
2. The property of bi-equivariance introduced to handle equivariance induced in both grasping and placing is an important and novel formulation (besides the fact that it was first introduced in an earlier paper by a subset of the authors, [20]) and was adequately illustrated in Figure 2-3.
3. The model is designed with many great properties that could be very important for efficient visual robotic manipulation learning. The use of generative modeling enables the method to handle multi-modality more gracefully compared to existing approaches, like [23]. The incorporation of the locality is well-motivated and could greatly provide a graceful alternative for the need for scene segmentation, and sources of inspiration for locality are well-cited.
4. The use of a query density model enables the method to be end-to-end trainable, and thus nicely eliminates the need for human-annotated/heuristically sampled query keypoints, as used in NDF.
5. The work adequately addressed limitations and directions for future works in Section V.

**Weaknesses**
1. The paper was densely written and the writing felt rushed, missing some minor notational definitions. For example, $p$ in the final sentence of section II is not defined.
2. The choice of using Wigner D-matrices was not clearly motivated, besides the fact that they are “irreducible representations of the SO(3) group”. While it’s understandable that the workshop paper page limit is rather short, perhaps more justification for choosing Wigner D-matrices over other rotation representations could be added to the appendix or a longer version of the paper.
3. The function of color, $c_i$ as introduced in section II, is not clearly explained, why does it need to be included as the point cloud representation, $O$, in addition to the position $x_i$? Another mention of this seems to be in Section III.C, but the part on “color vector” was very confusing and seemed out of the blue.
4. All experiments were done in simulation only, the applicability of the proposed method to real-world physical manipulation tasks remains unknown. Especially, given that the model seems to require both the position and color of points in point cloud observation, the sim2real difference between captured point clouds in the simulation from the ones captured from the real world, in terms of both sensor noises and non-idealities in captured position and color should be discussed in writing, if not further analyzed via real-world experimentation.
5. The paper could benefit from empirically comparing with more baseline methods; and if not, additional detailed explanations of why certain baselines are included, while not others. For example, it’s not clear why only NDF [23] was adapted with descriptor vectors as comparing baseline, instead of the more recent L-NDF [3] or R-NDF [24], which according to Table I, are more similar to the proposed methods in terms of the five factors the table listed.

---

### Decision · Program_Chairs · 2023-06-24

**Decision:**

Accept (Oral)

**Comment:**

Congratulations! We encourage the authors to revise the paper based on the reviewer's feedback.
Your paper will be presented as both a long oral presentation and a poster. Detailed instructions about the presentation format and camera-ready submission will be sent to you soon.